# Association between the Preoperative Standard Uptake Value (SUV) and Survival Outcomes after Robotic-Assisted Segmentectomy for Resectable Non-Small Cell Lung Cancer (NSCLC)

**DOI:** 10.3390/cancers15225379

**Published:** 2023-11-12

**Authors:** Aihab Aboukheir Aboukheir, Emilio Q. Villanueva, Joseph R. Garrett, Carla C. Moodie, Jenna R. Tew, Eric M. Toloza, Jacques P. Fontaine, Jobelle J. A. R. Baldonado

**Affiliations:** 1Department of General Surgery, Saint Luke Episcopal Medical Center, General Surgery Residency, Ponce Health Sciences University, Ponce, PR 00716, USA; aihab.aboukheiraboukheir@moffitt.org; 2Department of Thoracic Oncology, Moffitt Cancer Center, Tampa, FL 33612, USA; joe.garrett@moffitt.org (J.R.G.); carla.moodie@moffitt.org (C.C.M.); jenna.tew@moffitt.org (J.R.T.); eric.toloza@moffitt.org (E.M.T.); jacques.fontaine@moffitt.org (J.P.F.); 3Department of Pathology, University of the Philippines College of Medicine, Manila City 1000, Philippines; eqvillanueva@up.edu.ph; 4Expanded Hospital Research Office, UP–Philippine General Hospital, Manila City 1000, Philippines

**Keywords:** non-small cell lung cancer (NSCLC), segmentectomy, standard update value (SUV), lobectomy, overall survival (OS), recurrence-free survival (RFS)

## Abstract

**Simple Summary:**

Early-stage lung cancers are best treated with lung resections such as segmentectomy. Despite curative lung resections, however, lung cancers can recur. In our study we present our outcomes for robotic segmentectomy performed for lung cancer and how outcomes could be related to standard uptake value (SUV) as reported in the patients’ PET CT scan. We show here that robotic segmentectomy is safe and feasible for early-stage lung cancer, however recurrence was 28.4%. A higher preoperative SUV was associated with worse pathology outcomes and higher recurrence. A higher preoperative SUV was also associated with better long-term survival outcomes.

**Abstract:**

Background: Lung-sparing procedures, specifically segmentectomies and wedge resections, have increased over the years to treat early-stage non-small cell lung cancer (NSCLC). We investigate here the perioperative and long-term outcomes of patients who underwent robotic-assisted segmentectomy (RAS) at an NCI-designated cancer center and aim to show associations between the preoperative standard update value (SUV) to tumor stage, recurrence patterns, and overall survival. Methods: A retrospective analysis was performed on 166 consecutive patients who underwent RAS at a single institution from 2010 to 2021. Of this number, 121 robotic-assisted segmentectomies were performed for primary NSCLC, and a total of 101 patients were evaluated with a PET-CT scan. The SUV from the primary tumor was determined from the PET-CT. The clinical, surgical, and pathologic profiles and perioperative outcomes were summarized via descriptive statistics. Numerical variables were described as the median and interquartile range because all numerical variables were not normally distributed as assessed by the Shapiro–Wilk test of normality. Categorical variables were described as the count and proportion. Chi-square or Fisher’s exact test was used for association. The main outcomes were overall survival (OS) and recurrence-free survival (RFS). Kaplan–Meier (KM) curves were constructed to visualize the OS and RFS, which were also stratified according to tumor histology, the pathologic stage, and standard uptake value. A log-rank test for the equality of survival curves was performed to determine significant differences between groups. Results: The most common postoperative complications were atrial fibrillation (8.8%, 9/102), persistent air leak (7.84%, 8/102), and pneumonia (4.9%, 5/102). The median operative duration was 168.5 min (IQR 59), while the median estimated blood loss was 50 mL (IQR 125). The conversion rate to thoracotomy in this cohort was 3.9% (4/102). Intraoperative complications occurred in 2.9% (3/102). The median hospital length of stay was 3 days (IQR 3). The median chest tube duration was 3 days (IQR 2), but 4.9% (5/102) of patients were sent home with a chest tube. The recurrence for this cohort was 28.4% (29/102). The time to recurrence was 353 days (IQR 504), while the time to mortality was 505 days (IQR 761). The NSCLC patients were divided into the following two groups: low SUV (<5, *n* = 55) and high SUV (≥5, *n* = 47). Statistically significant associations were noted between SUV and the tumor histology (*p =* 0.019), tumor grade (*p =* 0.002), lymph-vascular invasion (*p =* 0.029), viscera-pleural invasion (*p =* 0.008), recurrence (*p* < 0.001) and the site of recurrence (*p =* 0.047). KM survival analysis showed significant differences in the curves for OS (log-rank *p*-value 0.0204) and RFS (log-rank *p*-value 0.0034) between the SUV groups. Conclusion: Robotic-assisted segmentectomy for NSCLC has reasonable perioperative and oncologic outcomes. Furthermore, we demonstrate here the prognostic implication of preoperative SUV to pathologic outcomes, recurrence-free survival, and overall survival.

## 1. Introduction 

Non-small cell lung cancer (NSCLC) is the leading cause of cancer mortality worldwide. There are many risk factors for developing NSCLC, the most notable of which include smoking history, environmental exposure, and family history. Once diagnosed and staged as early, surgical resection is the standard of care for treatment.

Anatomic pulmonary segmentectomy is gaining popularity among thoracic surgeons with an improvement in diagnostic imaging and surgical techniques using minimally invasive surgery. Historically, segmentectomy has not always been accepted by surgeons from the beginning due to the technical complexity and high risk of prolonged air leaks and the supposed increase in local recurrence in comparison with lobectomy [1]. 

There are two types of sublobar resection NSCLC: wedge resection and segmentectomy. The distinction between the two is that segmentectomy requires the oncologic standard of lobectomies, such as the anatomy of the separation of pulmonary segmental veins, arteries, bronchi, and the removal of lung parenchymal tissue [2]. The European Respiratory Society (ERS) and the European Society of Thoracic Surgery (ESTS) established clinical guidelines for sublobar resections, including wedge and segmental resections. They recommend that anatomical segmentectomy could be performed in the following scenario: stage 1A (tumor size 2–3 cm) with margins of resection > 1 cm, a stage I patient with poor lung functions and lung resections after prior lobectomy all of them with a level of evidence 2 grade of recommendation D [3].

Some argue that the long-term reduction (>12 months) in lung function induced by segmentectomy is very small in comparison with lobectomy but that small differences could benefit lung cancer patients who need subsequent lung resections [4]. Segmentectomy could also be offered to high-risk patients who otherwise could not tolerate a major pulmonary resection and are expected to become ventilator-dependent postoperatively [5,6]. Data regarding differences comparing sublobar resection vs. lobectomy have been abundant. Some studies establish that sublobar resections are associated with an increased incidence of local recurrence when compared with lobectomy [7]. Regarding postoperative complications, segmentectomy patients experienced a prolonged air leak, more blood loss, and increased operative duration [8], but no statistical significance was found in terms of perioperative mortality and morbidity between lobar and sublobar resections [9].

A more recent multicenter study found that with sublobar resection, there were better perioperative outcomes without compromising the short-term survival of elderly patients with clinical stage I NSCLC, but lobectomy was still selected if accurate lymph node staging was needed [10]. With the newer published non-inferiority trial, the JCOG0802/WJOG4607L for small lung tumors (diameter ≤ 2 cm, consolidation-to-tumor ratio > 0.5) demonstrated segmentectomy to be non-inferior to lobectomy with regard to overall survival and a reduction in forced expiratory volume in 1 sec (FEV1), despite a higher proportion of patients with local relapse in the segmentectomy group. This landmark trial suggested that segmentectomy should be the standard surgical procedure for stage 1 small-sized, peripheral NSCLC [11,12]. 

Robotic surgery (RS) has advanced and become popular in many fields; robotic lobectomy was first reported in the field of thoracic surgery and has become a common surgical procedure with acceptable outcomes in terms of operative time and blood loss [13]. Advancements in RS include high-definition three-dimensional videos, improved ergonomics, a less steep learning curve, tremor suppression, and better maneuverability of instruments, which can promote complex movements in a closed space and influence perioperative outcomes [14,15]. Although advancements in RS have helped in the field of lung surgery, segmentectomy is associated with technical challenges because it requires a deep hilar dissection to identify the segmental branches that need to be divided or preserved and the division of multiple intersegmental planes [16,17]. 

In the preoperative workup, a 2-deoxy-2-[^18^F] Fluro-D-glucose positron-emission tomography (^18^F-FDG PET/CT) is widely used in lung cancer for staging, operative planning, restaging, and the evaluation of the treatment response. The PET/CT scan provides clinicians with additional prognostic indications regarding survival and the estimated risk of relapse utilizing the assessment of the standardized uptake value (SUV) [18,19], which has also been demonstrated to be a parameter for locally advanced disease and poor survival in esophageal cancer [20]. It was found that diagnosis using PET/CT resulted in a substantial detection rate of postoperative lymph node metastasis pathologically, leading to poor prognosis despite the successful complete resection in clinical stage I lung cancer [21].

We report here the perioperative outcomes, including clinical and pathologic, as well as recurrence-free survival and overall survival outcomes in patients who had a robotic-assisted segmentectomy at an NCI-designated comprehensive cancer center. We also aim to show associations between the preoperative SUV and tumor stage, grade, and long-term outcomes.

## 2. Patient and Methods

We retrospectively analyzed patients who underwent a robotic-assisted segmentectomy (RAS) at a single institution from 2010 to 2021. The Moffitt Cancer Center (MCC) robotic surgery database was used for this analysis. This is a database that has been retrospectively collected and prospectively maintained under an IRB-approved protocol in the Thoracic Oncology Program. This study was conducted in accordance with the Declaration of Helsinki (as revised in 2013). Ethical approval to report this study was obtained from our institution’s Scientific Review Committee (MCC #16728, #18761, and #19304) and by our university’s Institutional Review Boards (USF IRB #Pro00022263 and Chesapeake IRB #Pro00017745 and #00000790). Individual consent for this retrospective analysis was waived.

Multiple variables, including clinical, surgical, and pathologic profiles, as well as perioperative outcomes of patients with primary lung malignancy who underwent RAS, were collected, and data were summarized using descriptive statistics. Only patients with primary NSCLC staged with a [^18^F]-fluorodeoxyglucose positron emission tomography/computed tomography (PET/CT) scan were included in the analysis. The preoperative SUVmax was determined from the primary tumor. 

Numerical variables were described as a median and interquartile range (IQR) because all numerical variables were not normally distributed, as assessed by the Shapiro–Wilk test of normality. Categorical variables were described as the count and proportion. Chi-square or Fisher’s exact test was used for association. The primary endpoints were overall survival (OS) and recurrence-free survival (RFS). Days-to-mortality and days-to-mortality/recurrence were right-censored to the date of their last known follow-up among those who did not die or did not have documented recurrence. Kaplan–Meier (KM) curves were used to demonstrate the OS and RFS of these patients, which were stratified according to tumor histology, pathologic stage, and preoperative SUV. In order to determine significant differences between groups, the log-rank test for equality of survival curves was performed.

## 3. Results

Between 2010 and 2021, 166 patients underwent RAS at MCC. Among these, 121 RASs were performed for primary early-stage NSCLC. The patients who had secondary malignancies, a history of prior primary NSCLC, and benign conditions were excluded. Of the 121 patients, 101 were evaluated with a PET-CT scan. Patient clinical characteristics are shown in Table 1. The median age was 72 years (IQR 12), and 60 patients (59.4%) were females. Most of our patients identified themselves as white (93/101, 92.1%) and were hypertensive (63/101, 62.4%). A left-sided segmentectomy was more common (66/101, 65.3%) than a right-sided one. The mean preoperative tumor size was 1.75 cm (IQR 1.2). A preoperative SUV ≥ 5 was identified in 46/101 patients (45.5%). 

Table 2 shows the pathologic profile of our patients. Adenocarcinoma was the most common (65/101, 64.4%), and squamous cell carcinoma was second (24/101, 23.8%). Only 12 (11.8%) patients had other pathologies. In terms of tumor grade, 50/100 (50.00%) were moderately differentiated, and the mean pathologic tumor size was 1.8 cm (IQR 1.1), 12/101 (11.9%) patients had lympho-vascular invasion, 23/101 (22.8%) had viscero-pleural invasion and the majority had stage I disease 82/101 (81.2%).

In Table 3, we demonstrate the surgical outcomes. The median estimated blood loss was 50 mL (IQR 125), while the median operative duration was 168.5 min (IQR 59). The conversion rate to thoracotomy in this cohort was 4.0% (4/101). Intraoperative complications occurred in 3.0% (3/101).

In Table 4, we further demonstrate our perioperative outcomes, including postoperative complications. The most common postoperative complications were atrial fibrillation (9.0%, 9/101), prolonged air leaks (8.0%, 8/101), pneumonia (5.0%, 5/101), and hypoxia (5.0%, 5/101). The thirty-day mortality rate was 0.0%. The median hospital length of stay was 3 days (IQR 3). The median chest tube duration was 3 days (IQR 2), but 5.0% (5/101) of patients were sent home with a chest tube. Recurrence for this cohort was at 27.8% (28/101). Time to recurrence was 353 days (IQR 504), while time to mortality was 505 days (IQR 761).

Table 5 demonstrates the association of SUV with pathologic factors. The 101 patients were divided into the following 2 groups: low SUV (<5, *n* = 55) and high SUV (≥5, *n* = 46). There were statistically significant associations between SUV and tumor histology (*p* = 0.016), tumor grade (*p* = 0.002), tumor size (*p* = 0.001), lympho-vascular invasion (*p* = 0.029), visceral-pleural invasion (*p* = 0.008), recurrence (*p* < 0.001) and the site of recurrence (*p* = 0.047). Interestingly, the pathologic stage and nodal metastasis were not associated with preoperative SUV in this cohort of patients. 

Kaplan–Meier survival plots (Figure 1 and Figure 2) showed significant differences in the curves for OS (log-rank *p*-value 0.0204) and RFS (log-rank *p*-value 0.0034) when stratifying for SUV uptake. The patients with a higher SUV uptake had a lower OS and RFS.

## 4. Discussion

This study presents the experience of a single institution with robotic-assisted segmentectomy to treat NSCLC and the value of preoperative SUV. Segmentectomy has historically not been utilized due to the suggested increase in perioperative complications like prolonged air leaks, increased estimated blood loss (EBL), and the increased duration of chest tube days. These findings have been consistently demonstrated to be of no statistical significance when compared to lobectomy [22]. At our institution, perioperative events are comparable to single-institution robotic lobectomies with a similar population, where EBL, chest tube duration, and operative time are similar [23]. Our LOS comes from the standardized management of all our patients with enhanced recovery pathways, which are comparable to the early discharge described by Chevrollier et al. [24]. Our experience with RAS demonstrates an acceptable operative time and blood loss with only ~3% intraoperative complications. Most postoperative complications were atrial fibrillation and pneumonia, which are comparable to the literature [13,25,26,27]. 

Unfortunately, despite a curative resection for early-stage NSCLC, there continues to be a high rate of recurrence, which currently ranges from 30 to 60% [28]. The recurrence rate in our cohort is consistent with this at 28.4%. We hypothesized in our study that higher preoperative SUV could be a poor risk factor for pathologic outcomes as well as long-term survival outcomes, including recurrence and OS. In addition to its role in diagnosis and metastatic workups for NSCLC, a meta-analysis confirmed that the increased SUV of the primary tumor is a poor prognostic factor in patients with NSCLC, with a combined hazard ratio of 2.27 [29]. A retrospective review performed for NSCLC tumors ≥ 1 cm found that increasing tumor size was an independent predictor of a higher SUV_max_ and that a high SUV_max_ to tumor size ratio is a stronger predictor of survival than a high SUV_max_ alone [30]. Another study that looked at preoperative SUV in patients who underwent a lobectomy demonstrated that high SUVmax was associated with larger tumor size, poor differentiation, lymphovascular invasion, and shorter freedom from recurrence [31]. All these studies support the prognostic value that preoperative SUV_max_ affords. 

We demonstrate here the value of preoperative SUV in relation to postoperative pathologic factors, showing that with a higher SUV, there is worse overall survival and worse recurrence-free survival. The standard uptake value has been an important determinant in lung cancer, with studies placing it as a risk factor for recurrence in small-sized early-stage NSCLC [32]. Both SUVmax and SUVmean values were also noted in one retrospective study to be significantly higher in those with lymph node metastasis compared to those without [33]. Kamigaichi et al. studied the predictive criteria of unexpected N2 disease findings in patients with preoperative SUV ≥ 3 and regarded it as an independent risk factor [34]. Likewise, higher SUVs were associated with worse survival for patients undergoing surgery and a marginal risk factor for OS [18]. 

Patients with preoperative SUV ≥ 5 in our cohort presented statistically significant associations in terms of tumor histology, tumor grade, lymph vascular invasion, visceral-pleural invasion, recurrence, and the site of recurrence. Sun et al. studied 200 patients who had lobectomies and identified preoperative SUV < 2.5 to be one of the prognostic variables for lower lymph node metastasis and cancers that could be treated with a segmentectomy [35]. This study showed that the probability of isolated lymph node metastasis in patients with a baseline SUVmax > 2.5 was 9 times that of patients whose SUVmax did not exceed 2.5, and concluded that the lower the SUVmax, the lower the chance of metastasis. All these data, including ours, suggest that a higher preoperative SUV portends a poorer prognosis. It could be a useful tool to predict lymph node metastases, recurrence patterns, and survival. More importantly, this prediction of tumor response and outcomes may offer clinical significance when optimizing treatment strategies. 

A meta-analysis of previous studies on stage I lung cancer also demonstrated that patients with tumors with a higher metabolic activity have shorter survival than patients with tumors with lower metabolic activity via evaluating lympho-vascular invasion, SUV, and disease-free survival [36]. Kaplan–Meier survival analysis in our study showed significant differences in the curves for OS and RFS between the SUV groups, with a higher SUV portending poorer OS and RFS. With the publication of the JCOG 0802 and CALGB 140503 results, we now have strong data to support segmentectomy as a non-inferior alternative to lobectomy in treating early-stage, ≤2 cm NSCLC. We expect that based on these results, there could be an increase in the number of patients with early-stage, resectable NSCLC who are treated with segmentectomy. These two trials did not investigate and stratify patients based on preoperative SUV; however, and what we now know based on our study findings and the literature, future studies could dwell on how preoperative SUV can play a role in further influencing treatment decisions, including induction and adjuvant therapies. 

## 5. Study Limitations

There are several limitations to this study. First, this is a retrospective analysis of prospectively collected data in a single-institution study only. In addition, there are no clear criteria as to the performance of robotic-assisted segmentectomy versus lobectomy for the population of patients in this study. In addition, SUV is affected by many factors, such as differences in body part composition, time-dependent factors, and PET scanner calibration. It is only a single-pixel value. There is no standardized cut-off value to establish a relationship between OS and RFS. 

## 6. Conclusions

Robotic-assisted segmentectomy for NSCLC appears to have reasonable perioperative and oncologic outcomes. Furthermore, we demonstrate here the prognostic implication of how a higher preoperative SUV leads to higher recurrence, more lymph-vascular invasion, viscera-pleural invasion, worse overall survival, and recurrence-free survival. This study opens the discussion for taking SUV into consideration for the further treatment of stage 1 NSCLC postoperatively. 

## Figures and Tables

**Figure 1 cancers-15-05379-f001:**
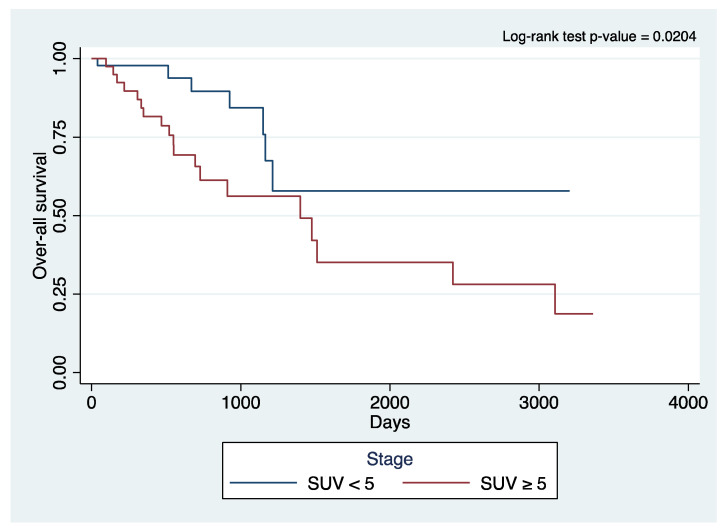
Kaplan–Meier curve of overall survival of primary lung cancer patients who underwent robotic surgery stratified to the standard uptake value.

**Figure 2 cancers-15-05379-f002:**
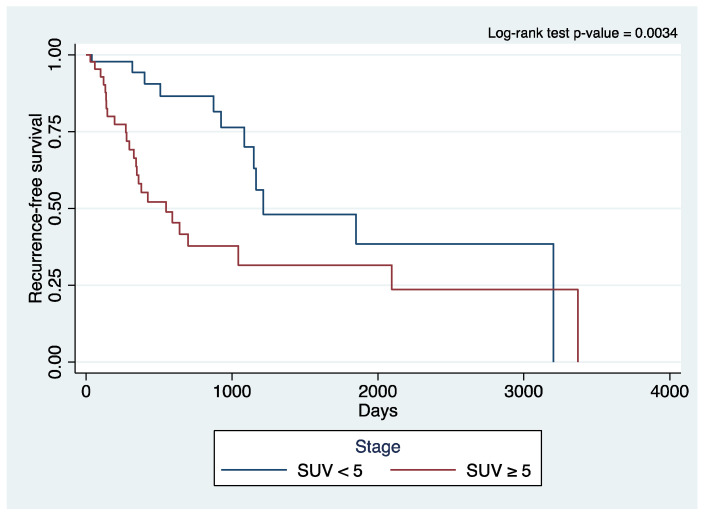
Kaplan–Meier curve of recurrence-free survival of primary lung cancer patients who underwent robotic surgery stratified to the standard uptake value.

**Table 1 cancers-15-05379-t001:** Clinical profile of primary lung cancer patients who underwent RAS.

Clinical Profile	Summary Statistics	
	*x ~/n*	*IQR/%*
Age	72	12
Sex		
Male	41	40.6%
Female	60	59.4%
BMI	27.1	7.1
Race		
White	93	92.1%
Hispanic	2	2.0%
Black	5	4.9%
Others	1	1.0%
Pre-operative FEV1	2	0.745
Pre-operative FEV1%	80.5	31.5
Co-morbidities		
Hypertension	63	62.4%
Diabetes mellitus	16	15.8%
COPD	42	41.6%
Congestive heart failure	4	4.0%
ESRD	0	-
Ever-smoker	58	57.4%
Chronic steroid use	1	1.0%
Laterality		
Left	66	65.3%
Right	35	34.7%
Tumor size, clinical imaging	1.75	1.2
Standard uptake value		
SUV < 5	55	54.5%
SUV ≥ 5	46	45.5%

**Table 2 cancers-15-05379-t002:** Pathologic profile of primary lung cancer patients who underwent RAS (*n* = 101).

Pathologic Profile	Summary Statistics	
	*x ~/n*	*IQR/%*
Tumor histology		
Adenocarcinoma	65	64.4%
Squamous	24	23.8%
Carcinoid/Neuroendocrine	11	10.9%
Other Lung Cancers	1	0.9%
Tumor grade *		
Well differentiated	25	25.0%
Moderately differentiated	50	50.0%
Poorly differentiated	25	25.0%
Tumor size, pathologic examination	1.8	1.1
Lymph-vascular space invasion **	12	11.9%
Viscero-pleural invasion **	23	22.8%
Positive margins of resection	1	1.0%
Total number of lymph nodes assessed	10	7
Number of mediastinal lymph nodes assessed	5	4
Positive lymph nodes	10	9.90%
Pathologic stage		
IA1	26	25.7%
IA2	30	29.7%
IA3	5	5.0%
IB	21	20.8%
IIA	4	4.0%
IIB	9	8.9%
IIIA	6	5.9%
IIIB	0	-
IIIC	0	-
IV	0	-

Note: * 2 cases were not graded, ** 1 case was not assessed for LVI and VPI.

**Table 3 cancers-15-05379-t003:** Surgical profile of primary lung cancer patients who underwent RAS.

Surgical Profile	Summary Statistics	
	*x ~/n*	*IQR/%*
Neoadjuvant therapy		
None	97	96.0%
Chemo/immunotherapy only	4	4.0%
Radiotherapy only	0	-
Combination chemoradiotherapy	0	-
Operative time	168.5	59
Estimated blood loss	50	125
Intraoperative complication	3	3.0%
Conversion to thoracotomy	4	4.0%

**Table 4 cancers-15-05379-t004:** Outcomes of primary lung cancer patients who underwent RAS.

Outcomes	Summary Statistics	
	*x ~/n*	*IQR/%*
Postoperative complications *	33	32.7%
Atrial fibrillation	9	9.0%
Prolonged air leak	8	8.0%
Pneumonia	5	5.0%
Pneumothorax	5	5.0%
Hypoxia	5	5.0%
Empyema	4	4.0%
Shock	3	3.0%
Aspiration	3	3.0%
Mucus plug	2	2.0%
Respiratory failure	2	2.0%
Other arrhythmia	2	2.0%
Chyle leak	2	2.0%
Myocardial infarction	1	1.0%
Cardiopulmonary arrest	1	1.0%
Hemothorax	0	-
Cerebrovascular accident	0	-
Pulmonary embolism	0	-
Length of hospital stay	3	3
Days with chest tube	3	2
Sent home with chest tube	5	4.90%
Days at home with chest tube	24	20
30-day mortality	0	-
Mortality	25	24.8%
Time-to-mortality	505.5	761
Recurrence	28	28.43%
Time-to-recurrence	353	504
Site of recurrence		
Nodal	4	13.79% ^
Pleural	2	6.90%
Local	10	34.48%
Distant	13	44.83%

Note: * Several patients had more than one postoperative complication.

**Table 5 cancers-15-05379-t005:** Association of SUV with the following pathologic factors among primary lung cancer patients who underwent robotic surgery.

Factor	SUV < 5	SUV ≥ 5	*p*-Value
	*n* = 55	*n* = 46	
	*n (%)*	*n (%)*	
Tumor histology			0.016
Adenocarcinoma	42 (64.6%)	23 (35.4%)	
Squamous cell	9 (37.5%)	15 (62.5%)	
Neuroendocrine	4 (36.3%)	8 (66.7%)	
Tumor grade			0.002
Well differentiated	19 (76.00%)	6 (24.00%)	
Moderately differentiated	29 (58.00%)	21 (42.00%)	
Poorly differentiated	7 (28.00%)	18 (72.00%)	
Pathologic stage			0.244
Early (stages I–II)	53 (55.79%)	42 (44.21%)	
Late (stages III–IV)	2 (33.3%)	4 (66.7%)	
Tumor size			0.001
T1	45 (67.16%)	22 (32.84%)	
T2	9 (33.33%)	18 (66.67%)	
T3	1 (14.29%)	6 (85.71%0	
Lymph node metastasis			0.506
N0	51 (56.04%)	40 (43.96%)	
N1 + N2	4 (40.00%)	6 (60.00)	
Lymph-vascular space invasion			0.029
With	3 (25.00%)	9 (75.00%)	
Without	52 (58.43%)	37 (41.57%)	
Viscero-pleural invasion			0.008
With	7 (30.43%)	16 (69.57%)	
Without	48 (61.54%)	30 (38.46%)	
Recurrence			<0.001
With	7 (25.0%)	21 (75.0%)	
Without	48 (65.8%)	25 (34.2%)	
Site of recurrence			0.047
Nodal	0	4 (100.0%)	
Pleural	1 (50.0%)	1 (50.0%)	
Local	5 (55.6%)	4 (44.4%)	
Distant	1 (7.7%)	12 (92.3%)	

## Data Availability

The authors confirm that the data supporting the findings of this study are available within the article. Raw data that support the findings of the study are available from the corresponding author upon reasonable request.

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
