# Peer review of "Association between the Preoperative Standard Uptake Value (SUV) and Survival Outcomes after Robotic-Assisted Segmentectomy for Resectable Non-Small Cell Lung Cancer (NSCLC)"

_cancers, 2023, doi:10.3390/cancers15225379_

Round 1
Reviewer 1 Report
Comments and Suggestions for Authors
Nice retrospective work which lacks a clear focus to my opinon.
If SUV is the major topic, why is blood loss, chest tube duration, etc important and how does it correlate to SUV? Maybe the focus should be altered to a single center experience of Robotic-assisted segmentectomy where the SUV may be an additional and interesting finding.
What LN stations are routinely assessed? This may be critical since it may influence prognosis. How was SUV assessed? What about controls, activity of FDG? Can SUVs from differnt PET-CTs be compared? I guess it is understood that a higher SUV may be a poor prognostic factor but can smaller variations be excluded? This may be important if used with cut-offs.
How does SUV correlate with tumor size?
Since SUV is correlated with differentiation grade and more nodal recurrence: What may be the consequences of assessing the SUV? Should it alter the operation procedure?
High SUV was not correlated with pathologic stage, different to other studies. What about lymph node metastasis (N1+N2 vs N0)?
In the discussion section: A hypothesis on why SUV is a poor risk factor should be added. Moreover: What should be the best cut-off since studies used different ones?
Since the last abstract is on the comparison segmentectomy vs lobectomy: How does SUV fit into this discussion?
Minor:
Standard uptake value: I guess it was determined from the primary tumor? Should be mentioned in the abstract as well.
Author Response
Nice retrospective work which lacks a clear focus to my opinion.
If SUV is the major topic, why is blood loss, chest tube duration, etc important and how does it correlate to SUV? Maybe the focus should be altered to a single center experience of Robotic-assisted segmentectomy where the SUV may be an additional and interesting finding.
Response: Blood loss, chest tube duration, length of stay, etc are important to support that robotic segmentectomy is a safe/viable option for early-stage NSCLC. It gives the readers an idea that this cohort of patients are consistent with national benchmarks, which would be important in the generalizability of results.
What LN stations are routinely assessed? This may be critical since it may influence prognosis. How was SUV assessed? What about controls, activity of FDG? Can SUVs from differnt PET-CTs be compared? I guess it is understood that a higher SUV may be a poor prognostic factor but can smaller variations be excluded? This may be important if used with cut-offs.
Response: For right-sided segmentectomies, lymph node stations assessed routinely are the following: 2R, 4R, 7, 8, 9, 10, 11. For left-sided resections, lymph node stations assessed routinely include 5, 6, 7, 8, 9, 10, 11. SUV in our study was assessed using the SUVmax as reported for the primary tumor in the official reading of PET/CT. SUV measurements are not sufficiently standardized between different PET-CT scans. A study by Doot et al for multiple scanners at multiple time points showed changes of around 15% (Doot R, Allberg K, Kinahan P. Errors in serial PET SUV measurements. J Nucl Med. 2010;51:126P.). One can expect the actual variability of SUVmax in practice to be greater than 15-20%.
How does SUV correlate with tumor size?
Tumor size is significantly associated with SUV. As the tumor size increases, the proportion of patients with SUV≥5 also increases..This has been added to the tables and results sections of the revised manuscript.
Since SUV is correlated with differentiation grade and more nodal recurrence: What may be the consequences of assessing the SUV? Should it alter the operation procedure?
The findings here that a higher SUV is correlated with tumor grade and nodal recurrence would lead to potential alterations with not only procedure but also institution of adjuvant or neoadjuvant strategies to treat NSCLC. Unfortunately, this is only a retrospective study and is not powered enough to make these recommendations. Findings are suggestive though of a higher potential for recurrence and hence makes the argument for preferring lobar rather than sublobar resections for cancers with high SUVmax.
High SUV was not correlated with pathologic stage, different to other studies. What about lymph node metastasis (N1+N2 vs N0)?
Added this new analysis, despite higher proportion of SUV≥5 among patients with lymph node metastasis (N1+N2) there is no sufficient evidence to conclude significant statistical association between lymph node metastasis and SUV.
In the discussion section: A hypothesis on why SUV is a poor risk factor should be added. Moreover: What should be the best cut-off since studies used different ones?
This is noted and has been addressed in the discussion. The cut-off values for defining high SUV varies a lot among different studies and different institutions, ranging from 2.5 up to 15. There has not been a study that defined the best cut-off value.
Since the last abstract is on the comparison segmentectomy vs lobectomy: How does SUV fit into this discussion?
Which abstract is this? I apologize if I do not understand the question. Our study only focuses on robotic segmentectomies. There are no robotic lobectomies in the population.
Minor:
Standard uptake value: I guess it was determined from the primary tumor? Should be mentioned in the abstract as well.
Yes, it was determined from the primary tumor. This has been addressed in both the abstract and in the methods section.
Reviewer 2 Report
Comments and Suggestions for Authors
I find this article almost flawless. The abstract section is too lengthy although (is there a recommended word limit for abstract?).
The authors might also want to discuss a few more articles in discussion (for example https://www.ncbi.nlm.nih.gov/pmc/articles/PMC8742841/ and others).
Also, a brief introduction focusing on etiology and pathogenesis of NSCLC would add value to the text.
Author Response
I find this article almost flawless. The abstract section is too lengthy although (is there a recommended word limit for abstract?).
Thank you for the comment. I have shortened the abstract.
The authors might also want to discuss a few more articles in discussion (for example https://www.ncbi.nlm.nih.gov/pmc/articles/PMC8742841/ and others).
Additional articles have been cited in the discussion, including the one mentioned above.
Also, a brief introduction focusing on etiology and pathogenesis of NSCLC would add value to the text.
This has been addressed on the revised manuscript as well, although I did not dwell on it too much.